# Human Rights and Religions: An Overview on a Controversial Relationship

Rafael Ruiz Andrés 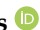

Department of Applied Sociology, Faculty of Political Sciences and Sociology, Complutense University of Madrid, 28223 Pozuelo de Alarcón, Spain; rafaelruizandres@ucm.es

**Abstract:** If Human Rights (HR) and religions are controversial concepts on their own, the relationship between the two can be fraught with complexity. However, the interest in jointly exploring the realities that underlie the two categories together has led to a long academic tradition of research on the relationship between religions and HR. In this article, based on a review of the state-of-the-art and from a post-secular perspective, we will approach the issue raised from two key points that will be in continuous interaction throughout the text. On the one hand, I will examine the interaction between religions and the polygenesis of HR, a vexed issue that has aroused growing attention in the academic community and that has marked the debate on the polyvalence of fundamental rights. On the other hand, I will analyze the studies that currently attempt to explore the potential contribution of religions to the promotion of HR in the present, but also pointing out the contradictions and ambiguities that arise in this endeavor. These two keys will show us the limits of the relationship between HR and religions, but also the opportunities that the explorations of the common substrate between the two offer cooperation between secular and religious actors in the joint promotion of Human Rights.

**Keywords:** Human Rights; religions; secularization; Axial Age; historical sociology; religious diversity; pluralism; modernity; post-secularism; sociology of religions

## 1. Introduction: The Debate on the Relationship between Two Controversial Terms

Human rights (HR) and religions are two widespread categories of organization of knowledge in the world. Although they are commonly associated and intertwined, both the multiple meanings of each and the relationship between them are complex and contested. We could easily start a conversation with a large part of the world's population on either of these two issues, but the chances that the dialogue will be plagued by nuances, debates, contradictions, and misunderstandings are high.

In this regard, one of the questions that helps to explain this contradictory situation is the often repeated Western imprint on both concepts. On the one hand, although we can explore precedents and genealogies in different cultures around the world, and we must underline their universalist vocation, HR has been contemplated in intrinsic connections with the Western worldview (Taylor 1999). On the other hand, although beliefs, rites, and myths are found in different cultures, the concept of "religion", as we tend to use it, constitutes a category that is clearly related to a Western-centric and Christian-centric prism (Fitzgerald 2015).

However, and beyond the criticism that we can rigorously make from an academia viewpoint, the reality of the expansion of both categories at a global level is evident. As Michael Ignatieff (2018, p. 26) points out, both the secular language of HR, through what Joas (2013, p. 174) calls a process of "value generalization", and religious languages, through the expansion of the secular–religious binomial to the different contexts of the world (Miller 2016, pp. 13–14), are part of the global culture.

The problem posed at the beginning of this text reaches a further level when we decide to link both HR and religions—and the myriad of different realities that underlie them—with the particle "and". What is the possible relationship between the two elements?

There is certainly a first, simpler level, which takes us into the presence of religious freedom in the very contents of the Universal Declaration of Human Rights (1948). In fact, this Charter, one of the most outstanding achievements in the history of HR that neither begins nor ends with this milestone, incorporates the rights relating to religion (Articles 2, 16, 18, and 26), which were subsequently extended to the Declaration on the Elimination of All Forms of Intolerance and of Discrimination Based on Religion or Belief, also known as the "1981 Declaration".

However, it is more complex to trace the history of interaction between HR and religious traditions (understood as the cultural repositories shaped by the historical trajectories of believers who have belonged to—or are included by subsequent hermeneutics in—a religious community), and to analyze how they have mutually conditioned each other. This endeavor has, nevertheless, been widely addressed in the history of intellectual and academic reflection. In this regard, we find a first division between those who point to a stronger or weaker influence of religions on the genesis of HR (Henkin 1998; in Breskaya et al. 2018, p. 423).

To this first dilemma, we can add another debate, long present in the academic tradition of the last century, about what kind of relationship is established between religions and HR, which is, in turn, framed within the broader controversy about the relationship between religions and modernity. In this sense, we would find those who, following the thesis of ex novo modernity (Blumenberg 2008, p. 27), would defend the total difference between religions and HR as parallel histories that, when they have intertwined, have done so in a conflictive manner. In contrast to this would be those who point out, in one way or another, that there is a relationship of continuity, also expressed in terms of substitution or translation, between the Christian tradition and HR. For the latter perspective, the Christian tradition would constitute the foundation without which HR could not be understood (Žižek 2002, pp. 156–57; Ferry and Gauchet 2007, p. 15; Morgan 2009, p. 3), since Christianity's reflections on human dignity would chronologically precede and thus theoretically form the basis from which Human Rights may have emerged.

Despite the undoubtable interest that these questions create, these debates have been deadlocked, as their answers are often intermingled with statements that may ultimately be grounded in the deeper consideration about what the foundations of HR should (or should not) be. In a way, these debates tend to reproduce two compact blocs, religions and secularities, which struggle for primacy and legitimacy in the genesis of "Western culture".

This long-standing controversy between HR and religions, and ultimately between modernity and religions, has been complemented by a whole series of debates that in recent years are shaping the perspective and approach to the relationship between HR and religions. The growing interest in the joint analysis of religions and Human Rights has also been encouraged by a series of events of varying depth and content, ranging from the global impact of 9/11, the visibility of interfaith dialogue initiatives, or the position of religions, sometimes as advocates and promoters, sometimes as opponents, of important debates on Human Rights. This has led to growing attention provided to the issue in disciplines such as international relations (Williams 2022) or sociology (Breskaya et al. 2018).

The aim of this article is to examine the relationship between religions and HR from the perspective raised by the academic reflection in recent decades, which have added nuance, ambiguity, and richness to the interaction between the two. To this end, I will adopt a position that I will describe as "post-secular", as the premises of this approach will be the framework for my stance on the question posed. Post-secularity will allow us to address the issue raised by the keys that have emerged in the growing debate of recent years, and which at the same time will allow us to establish a dialogue with the other articles included in the Special Issue of *Religions* "Public Management of Religious Diversity and Human Rights in Post-Secular Societies".

The debate on post-secularization has been developed over the last two decades by authors from different disciplinary backgrounds, but united by the same urgency to reflect on religious trajectories in the modern world, as well as the desire to investigate a genesis of secularization that, in contrast to some approaches to the process, is perceived as more complex, contradictory, and ambiguous (Ruiz Andrés 2022a). Along with the essential works of the philosopher Jürgen Habermas (2006, 2008, 2009, 2011), this category has also been adopted in the reflections of different prominent voices in the sociology of religion who—in addition—had previously theorized on the concept of secularization, for example, Bryan S. Turner (2010) and José Casanova (2017). Although post-secularization has had multiple interpreters and variants, I will focus on two ideas that will allow us to build our approach and, at the same time, to establish a dialogue with the developments in the debate on religions and HR.

On the one hand, post-secularization posits the need to deepen and strengthen awareness "of the complexity of secularization" (Turner 2010, p. 651), an effort that in turn connects with critical studies on genealogies of secularization (Asad 2003; Casanova 2012a). Applied to the current debate on HR and religions, this post-secular premise is linked to the various attempts to apply a hermeneutic (Witte and Green 2011) that can construct a genealogy of modernity and HR from the recognition of the porous boundaries between secularity and religions (Joas 2013, p. 4), and thus from the ambiguities and contradictions of the relations between religions and modernity in the genealogy of HR.

On the other hand, post-secularization values religions not merely as existing social facts, but as potential contributions to our societies (Ortega-Esquembre and García-Granero 2019, p. 73). Thus, it constitutes an analytical paradigm with particular interest in the present and future of our global society. With regard to the subject addressed by this article, the post-secular perspective prompts us to investigate what and how the role of religions in the public space and their contribution to democratic culture and HR should be. This post-secular approach, in turn, engages with the academic effort made in recent decades to delve into the relationship between HR and religion from empirical studies (Breskaya et al. 2018), as well as with those reflections that are committed to the study of HR as a point of support for interreligious and interconvictional dialogues (Gonçalves 2020; Council of Europe 2021).

Based on a review of the literature on HR and religions in recent decades and a post-secular approach that emphasizes the complex history of interactions between religions and secularity, this article will explore the relationship between religions and HR. To this end, we will discuss three fundamental ideas of the debate: the historical polygenesis and polyvalence of HR and their relationship with religions (Section 2), and the effective contribution of religions to the achievement of a culture of Human Rights in the present day (Section 3).

Firstly, in Section 2, we will explore the complexities of the religious genealogy of HR, brought to the fore in recent decades by debates such as the Axial Age, as well as the parallel developments of interreligious and interconvictional dialogue initiatives on the basis of the increasingly rediscovered universal and common principles. Around these questions, we will raise a conversation between the polyvalence and polygenesis of HR (Duranti 2012).

Secondly, in Section 3, we will critically examine a secularist view of modernity, highlighting how reflections in recent decades place a strong emphasis on the potential role of religions not only in the polygenesis and in polyvalence of HR but also in the actual defense and promotion of a democratic and HR-based culture at the global level.

In the conclusions, we will look at the possibilities and limits of this socio-historical perspective on the relations between religions and HR. The contradictions that have emerged allow for a more nuanced analysis of the debate and also offer multiple opportunities for the gestation of narratives that promote the polyvalence of HR and their understanding as a shared task in the present between religious and secular actors.

## 2. The Religious Roots of Human Rights: Debates and Opportunities on the Genealogy of Universal Principles

The United Nations (UN) Commission on Human Rights, which was responsible for drafting the 1948 Declaration, included people of different religious denominations, as well as representation from countries of various religious backgrounds and expert opinion from a wide range of cultural and religious contexts (Witte and Green 2011, p. 6). As Hans Joas (2013, p. 185) points out, this was key to the possibility that the values contained in the charter could be generalized and aim for universalization. The Declaration certainly does not include any reflection on any religious basis in its rationale[1]. However, and in addition to the above-mentioned religiously pluralistic composition of the Commission, the religious question was behind the debates that led to the Charter (Glendon 2001), and the respect for freedom of religious belief, as well as all the rights associated with it were included in the Declaration.

Prior to the Declaration, there had been a long history of relationship between religions and HR, and in particular debates on natural law and Christianity. In this respect, as Jacques Maritain (1948), one of the people involved in the Commission on Human Rights, pointed out, it is worth noting what he saw as a historical unfolding of the debates on natural law. In this concept, the French thinker underpinned the emergence of Human Rights, although not without contradictions, since he pointed out the distortion of the idea of natural law during modernity as well as the difficulties of reaching a universally shared and unquestioned consensus. Certainly, prominent voices and thinkers from the Christian tradition were part of the path towards HR (Taylor 1999; Williams 2022). Nevertheless, Christianity was also a source of criticism for the attempt to construct a list of rights not necessarily based on the divine foundation, not to mention the relative lack of response from the Vatican to the Declaration until the arrival of Pope John XXIII, who gave greater importance and relevance to HR as part of the foundations of his discourse in the complex times in which he lived (Castillo 2007). For some religious sectors, not only Christians, the HR were based on modern individualism (King 2011), which led to partial or total rejection among certain religious groups, as can be seen in the kind of "rectification" of the United Nations Declaration of Human Rights in the Declaration of Human Rights in Islam (DDHI), also known as the Cairo Declaration (1990), which threatened to produce a rupture in the intercultural consensus achieved by the 1948 Declaration.

Beyond these controversies about the role of religions in Human Rights, which will be present throughout the paper, the milestone of the 1948 UN Declaration of Human Rights can be read in continuation with the footsteps of the "*Etsi Deus Non Daretur*" ("As if God did not exist") of the 17th century Dutch jurist Hugo Grotius and the political and intellectual legacy of the 18th century, from which emerges the history of the secular sacralization of the person, which Joas (2013) sees as the basis of the culture of HR. At the same time, however, this willingness to represent different religious confessions and spiritual sensibilities was to a certain extent a sign of the recognition at least of the polyvalence of HR (Duranti 2012).

The academic analyses of recent decades on the issue have reinforced this double idea of independence and multiple relationship between secularity and religions in the genesis of HR, as shown by the milestone of 1948. In this sense, and beyond the historical debate between Christianity and modernity, there has been an ongoing effort to reconstruct a more complex, plural genealogy of HR, in which the boundaries between secular and religious are porous and, equally, in which the subject "religions" is conjugated in the plural.

From this willingness to explore the multiple and diverse roots of HR, the search for a genealogy of HR in religions has been strengthened, with a particular focus on the so-called World Religions for two reasons. On the one hand, because of the importance that the neo-Weberian paradigm of World Religions, with its possibilities and inadequacies, has had in religious studies during the beginning of the 21st century (Cotter and Robertson 2016). On the other hand, due to the expansion and popularity of the Axial Age debate, in which the search for the common roots of the various major religious traditions—as well as the possibilities of dialogue between them and modernity has been on the agenda in

recent years. Authors such as Robert Bellah, Hans Joas, and Shmuel Eisenstadt have taken up the debate on the Axial Age, a concept coined by Karl Jaspers (1985), in his book *Origin and Goal of History* (1934), to delve into the great revolution that took place in the middle of the first millennium before the Common Era. In this period, which Jaspers places between 800 BC and 200 BC, significant changes in the ideas of the individual, transcendence, *ethos*, and society—which would form the substratum of the world religions—took place in various contexts of our world. This can be evidenced by their appearance in part of the core texts of different religious traditions—"the canonical Hebrew prophets, Amos, Isaiah and Jeremiah, among others; [...] the Analects of Confucius and the Daodejing (perhaps the most translated text in the world); and early Indian texts such as the Bhagavadgita and the teachings of Buddha in the Pali Canon" (Bellah and Joas 2012)—which were also being composed during this Axial Age. A milestone in the study of the Axial Age was the conference "The Axial Age and its Consequences for Subsequent History and the Present"—held at the Max Weber Centre of the University of Erfurt in Germany, 3–5 July 2008—from which The Axial Age and its Consequences, edited by Bellah and Joas (2012), were derived.

One of the keys that has facilitated the connection between Axial Age studies and the genesis of HR is to be found in what the scholars point to as a potential universalization of values that occurred in various global contexts during this period. As Eisenstadt (2012, p. 278) states, during the Axial period, religious beliefs broke away from their attachment to the homeland and embraced universal values. This universalist substratum of the Axial Age would form the basis even of religions such as Christianity or Islam, which appeared later than the Axial Age, but which are also considered to share the axial background. It is precisely in the universalization of values that different researchers have sought to identify the eventual religious contribution to a polygenesis of HR, which also allow for polyvalence in their interpretation.

In the case of Judaism, although there has traditionally been a strong link in the interpretation between religious ethics—God–human relations, inter-human relations, and the relations of the individual with the community (Novak 2011, p. 30)—and the idea of peoplehood, the Biblical texts that appeared during the Axial Age, and especially the so-called prophetic books, would reinforce the universal factor in the interpretive possibilities of Judaism. The Axial period would enhance an universalizable potential in Judaism which, as Novak (2011, p. 33) points out, is also to be found in the exegesis of tradition by thinkers such as Maimonides.

The relationship between Christianity and HR is possibly the most studied in recent decades, due to the aforementioned nexus between Christianity and modernity, as well as the debates on natural law and Christianity, and their potential contribution for the creation of a fertile soil for Human Rights. Indeed, despite the complex history of the relationship between Christianity and HR, the authors also underline the universalizing potential of Gospel texts such as the parable of the Good Samaritan (Taylor 2015) and the interpretation from this perspective by the so-called Church Fathers (Wolterstorff 2011, pp. 52–53), which would later give rise to different debates on human dignity, as can be seen in the discussions on the status of indigenous people during the Spanish colonization of the Americas, with important milestones in history of HR such as the "Valladolid controversy" (1550–1551) and the reflections of the School of Salamanca (García y García 2008).

As for Islam, scholarly reflection has emphasized the universal vocation—to all humanity—of the Qur'anic text (7:159). In this regard, Sachedina (2009) not only points out the importance of seeking an Islamic foundation for HR, but also affirms that—from the dialogue between reason and revelation, and from Islamic sources, specifically from an exegetical reading of the Qur'an—it is possible to find elements to establish HR and pluralism in both the social and religious spheres (Sepúlveda del Río 2018, p. 626). Similarly, An-Na'im (2011, p. 67) argues that Sharia can be read through an exercise of "reframing" at the service of HR beyond the rights and jurisprudence of states, which is how it has usually been interpreted.

However, the Axial Age background is not confined to the monotheistic Abrahamic traditions, but also emerges—according to the researchers who have adopted this framework—in other regions of the world, particularly in East Asia and India. For this reason, the debate on the Axial Age has also expanded to the study of the socio-historical development of these contexts, as well as the search for universalizable principles in the religious traditions of this area that could also form part of this dyad of polygenesis and polyvalence of HR.

This is the case of Hinduism, which is a category created from a Western colonial perspective to refer to the set of Indian traditions that possessed an air of familiarity in rites and beliefs. In this regard, it is worth highlighting Menski's (2011, p. 81) reflection on Hinduism and HR. In addition to going through all those elements that are more problematic for the interaction between the two (for example the *Varṇa*, popularly known as the caste system), he also considers that there are certain principles that can be read from this potential for universalization since the ancient times of Vedic literature. "Hinduism does not intrinsically privilege Hindus, let alone Brahmins, males or anyone. Every created being, including animals, living organisms, and plants (as many vegetarians believe) are interconnected in a giant cosmic spider web, called "Hindu" much later in time, but the net itself is ancient".

Finally, in Buddhism, which is another major World Religion and is born from the substratum of Indian traditions, the ideas of equality and non-violence—defended in the words of the Buddha—would also be in potential interaction with this idea of universalizable values (King 2011, p. 108).

Despite the possibilities that this debate on the genealogy of HR and the spread of universalist values during the Axial Age has opened up in recent decades, we cannot ignore its many limitations, particularly three, which have also been highlighted by different studies.

Firstly, the lack of historical analyses to back up considerations and reflections on the Axial Age, which have been mainly carried out by sociologists and philosophers (Assmann 2012). In this respect, the hermeneutic possibilities offered by the Axial Age are also challenged by its attempt to establish shared frameworks across very different cultures and times, with the risk that hermeneutics may ultimately become a projection of present concerns onto the past. At this point, it is worth asking whether the polygenesis of Human Rights supports their polyvalence, or rather if it is a search for a historical basis to justify their polyvalence in the present day.

Secondly, while we can indeed see in this drive for universalization a part of the polygenesis or at least a common ground from which HR can be discussed, it is no less true that this universalizable potential of religions has also served to justify imperial projects, in which universalistic religious premises were imposed through violence, coercion, and elimination (Martin 2012, p. 194). Even today, many religious organizations are behind the promotion of Human Rights, but we cannot forget the multiple problems and dangers posed to HR by some approaches that also seek their justification in religious traditions.

Thirdly, and due to the usual restriction of the Axial Age to the so-called World Religions, it is worth noting the lack of analysis of this universalizable potential in other religious traditions, as Juan José Tamayo has recently shown in his study of the roots of compassion not only in the major religions but also in other traditions such as the Bahá'í and Amerindian peoples (Tamayo 2021). In this sense, we can affirm that although the plurality of the genealogies of HR and modernity has certainly been emphasized in other faiths beyond Christianity, it remains limited in the face of the super-diversity of religious and non-religious positions that populate the world today (Beaman 2017).

For these reasons, some of the researchers who have delved into these roots have also emphasized the risks involved. In particular, as Joas (2013, p. 10) points out, the risk that studies on the shared roots between religions and HR may be limited to a selection of cleverly chosen quotations that show the possibility of dialogue between religions and the foundations of HR, while concealing a longer, contradictory, and ambiguous history in this regard (Joas 2013, p. 140).

However, without ignoring the complexities and contradictions of this search for the religious genealogy of HR, we need to underline that, beyond the academic debate, the reflection on shared and universalizable principles has contributed to the normative justification of renewed readings on the legacy of religions and their potential for dialogue with other religious faiths and with secularity. In other words, we can say in a way that the very attempt to seek a polygenesis of HR has reinforced their polyvalence in parallel. Despite the ambivalences of the relationship between HR and religions, the simple recognition of this relational ambiguity at the historical level and the possibility of searching for joint semantic contents has fostered important dialogical initiatives in which HR have become a point of encounter between traditions and convictions (Gonçalves 2020).

In this sense, theologians such as Metz and Küng have opted for the task of building a fundamental ethic common to all religions, "which, despite the differences, takes the form of a consensus on the fact that the other has an inalienable value and possesses an inalienable dignity from which fundamental Human Rights derive" (González-Anleo 2008, p. 106). This proposal has borne an outstanding fruit in the declaration of a "world ethic" by the Parliament of World Religions (1993) (Aragão and Fernandes Cabral de Souza 2020).

Another initiative of interest has been led by the academic Karen Armstrong, who has devoted herself throughout her academic career to the search for the common roots of religions (Armstrong 1995, 2020). Following the Ted-Talk given by this leading scholar on 28 February 2008, the Charter of Compassion was published, a declaration that articulates and promotes initiatives around the search for these common and universalizable principles, particularly those related to "the shared values of empathy, kindness, justice, and interconnectedness", explicitly alluding to the defense and expansion of HR among its aims[2].

Equally noteworthy was the signing of the "Document on human brotherhood. For world peace and common coexistence", in a meeting between Pope Francis and Ahmad Al-Tayyeb, Grand Imam of Al-Azhar, which took place in Abu Dhabi in February 2019. From this shared and common genealogy between religions and HR, the Declaration underlines its commitment to the consolidation of HR, based on a justification that combines the theological backgrounds of the two religions involved in the document with universalizable principles. Between the two foundations (secular and religious), a sort of reference to natural law theoretically underlies both the universalist principles of the two religions and the basic notions of HR (Sachedina 2009). Thus, the mention of international law (point f) is interspersed with references to religious tradition through the appearance of a quotation from Sura 5 (32) ("whoever saves one life, it is as if he had saved the lives of all humanity"), or the use of the terminology of "widows and orphans", which ultimately refers to a metaphor present in the tradition of the Abrahamic religions to particularize the vulnerable sectors of any society (Ruiz Andrés 2022b).

Beyond its academic and scientific limitations and the multiple contradictions of the religious polygenesis of HR, this dialogue between religions and HR, whose theoretical exploration has also been encouraged in parallel, has not only allowed the consolidation of a common substratum of interaction between religions but has also favored dialogue between secularity and religiosity around the polyvalence of Human Rights. In this respect, Panikkar (2003) recalls that the search for common and universalizable languages between the different religions, as is necessary in the case of interreligious dialogue, simultaneously fosters the encounter with secularity (Walhof 2006, p. 585).

## 3. Beyond Secularism: The Contribution of Religions in Achieving a Global Culture of Human Rights

In his book *The Sacredness of the Person: A New Genealogy of Human Rights*, Hans Joas (2013) provides us with an affirmative genealogy of HR, which is rooted in a "history of the sacralization of the person" that is forged prominently from the eighteenth century onwards, as the author points out. However, at the beginning of his work, Joas presents other narratives that have competed for the interpretation of the genealogy of HR. Among

them, there is one that has its roots in the French Revolution and portrays HR as an alternative and even confrontational path to the religious tradition (Joas 2013, p. 4).

Certainly, this has been a common discursivity that was consolidated from the Enlightenment and the rejection during that period of violence and religious wars that occurred during the early modern period (Taylor 2015, p. 615). From this perspective, modernity, and all its achievements, such as HR, would appear as emancipation and in opposition to religious traditions. In parallel, there would be the forging of what José Casanova (2012b, pp. 100–1) calls a stadial-secularist consciousness, characterized, among other things, by the condition of having "overcome" the alleged irrationality of belief. In contrast to post-secularization, which would be the non-secularist alternative interpretation of secularization (Beck 2009, p. 90), this stadial-secularist consciousness would be the heir of a discursivity that is still present in parts of our world, particularly in Western Europe, and which tends to see the radical incompatibility of religions with modernity and secularity, thus to deny the effective contribution of religions to HR culture. This brings us to another of the major debates that, along with the questions raised about polygenesis and polyvalence, have repeatedly come up in the debate: can religions contribute to the promotion of a Human Rights culture?

In this sense, significant groups of Western European societies continue to link religion with violence. In the BBVA Foundation's European Values Survey 2019 (Fundación BBVA 2019), which examines a broad set of values and attitudes of the adult population in five European countries (Germany, the United Kingdom, France, Italy, and Spain), the predominant view is that "Religions today are more a source of conflict than of peace"[3]. Moreover, considerable percentages of the European population think that the teachings of some religions promote violence, a spectrum that ranges from 16 percent of Swedes supporting this statement to 33 percent of Italians (Pew Research Center 2018, p. 69).

At the academic level, the book *The Nonreligious. Understanding Secular People and Societies* (Zuckerman et al. 2016) argues that against the myth that religion promotes ethics and good behavior (synthesized in Dostoevsky's famous phrase "If God does not exist, everything is permitted"), it is the most secularized regions of the world that have the highest and best security levels.

In addition, it should not be forgotten that some of the performances of religions in the public sphere would encourage this stadial-secularist perspective. Along with the violent acts carried out in the name of religions, which have sometimes had the silence of some religious leaders, we cannot ignore the controversial positioning of certain religious sectors and prominent voices on HR at the beginning of the 21st century. This has led religions to sometimes engage in what Vuola (2005) calls fundamentalist ecumenical fronts in the face of the debates and claims of secular society, undermining their credibility as essential contributors to the history, present and future of HR.

Nevertheless, as Øistein Endsjø (2020) reminds us in his study on HR dialogue between religious traditions and LGTBIQ+ groups, just as it is naïve to try to see the entire history of religions as a fast track to Human Rights, it is equally simplistic to reduce the complexity of religious traditions to some positions of what may be minority but highly mobilized sectors. In fact, the ambiguity that we underlined about religions with respect to HR is equally extensible to these discussions, in which we find not only how the debates of civil society are reproduced within religious communities, but also we can also discover prominent religious voices in defense of all possible positions.

For this reason, and even though events such as those described may have reinforced stadial-secularist consciousness (Ruiz Andrés 2023), the denial of the potential contribution of religions to democratic society and thus their role in the genealogy and present of HR is to be seen in the light of a secularist interpretation of modernity, which for Habermas (2009, pp. 77–78) is equally as unhelpful to HR as religious fundamentalism. As Habermas points out (in Beck 2009, p. 162), "Secularism, if absolutized, entangles itself in a universalist circularity that blinds itself and us to the cosmopolitan recognition of the otherness of other confessional traditions".

Faced with this secularist view of modernity and HR, in recent decades, not only has the search for a more complex and broader genealogy of HR taken hold, but also the analysis of how religions can contribute to the consolidation of democracies and the defense of HR, a perspective that has reached its epitome in the post-secular perspective. In this sense, post-secularization is not only based on the discovery of a consciousness that is capable of revisiting the genealogy of modernity and HR (Mate 2013, p. 218), but also has a normative project based on the construction of a framework in which religions are recognized as essential actors in the defense of HR and democracy, and can effectively make their contribution from their heritage and repository of meanings. From this post-secular approach, different works have highlighted the multiple ways in which religions and their deposits of values could enhance HR, without in any case constituting a substitute for them, but rather as a support for democratic culture and HR (de Vries 2006, p. 49).

Habermas (2008) has reiterated two important contributions of religions in democratic societies. The first of these would revolve around the semantic potential of religious traditions, which, beyond the debate on the polygenesis of HR, can be translated into social values, political visions and motives of solidarity that have not yet found politically appropriate secular expression (Habermas 2011, p. 139).

Secondly, Habermas stresses the "emotional potential" that religions possess to ensure that citizens can commit themselves to communitarian values, which requires an affective and effective solidarity that cannot be coercively imposed (Panea Márquez 2009, p. 64).

Other authors have highlighted the contribution of religions to the defense, achievement, and expansion of HR, from the conviction that "intercultural discourses on the foundations of a more just international order can no longer be made unilaterally by the state sphere" (Habermas 2011, p. 128).

At the everyday context, where, as Ignatieff (2018, p. 177) points out, HR really begin, we cannot overlook the fact that despite continued secularization at the global level (Zuckerman et al. 2016; Inglehart 2021), the majority of our world's population continues to self-identify as religious. Moreover, in many contexts, large sectors of the population links ethics to religion. As the Pew Research study *The Global God Divide* points out, significant majorities in emerging economies connect morality to the idea of God, with percentages reaching over 70% in countries such as Brazil, Tunisia, India, Turkey, the Philippines, Indonesia, Nigeria, Kenya, South Africa, and Lebanon (Pew Research Center 2020). Given the evidence of the presence of religions in the everyday decisions and in the sources of morality of important groups around the world, religious faiths can play a fundamental role in the expansion of HR in the daily lives of populations for whom, perhaps, the culture of Human Rights "has not permeated local customs and continues to be part of the elitist discourse of activists, academics and legal experts" (Ignatieff 2018, p. 7).

At the level of state and societal articulation, authors such as Witte and Green (2011, p. 16) have highlighted that without religions and other civil society actors, the state can come to play a unique and exaggerated role in the protection of HR. In a certain way, the fact that the different state institutions can count on other stakeholders for the promotion of the culture of HR helps to overcome "the rigidities of procedural liberalism" and generate a Human Rights culture that is more adapted to a multicultural context (Taylor 2009, p. 99).

At the international scale—and from this multicultural perspective—religions can have an equally important contribution to make. As we pointed out earlier with Taylor (1999), one of the issues that has usually been criticized about HR is their supposedly Western character, which for the Canadian philosopher would be evident in the philosophical and conceptual architecture of HR. Because of this Western "appearance" that hangs over HR, their scope can be seen as limited and can sometimes generate rejection from sectors that see HR as a Western neo-colonial instrument and in contradiction with their cultural or religious values (An-Na'im 2011, p. 67). The dialogue between HR and their potential genealogy in the different religions of our world, particularly explored by the Axial Age debate, increases the possibility that religions can become bridges between the formulation of the Declaration of Human Rights and the traditions and semantic potentials of the

different cultures of our world, thus contributing to the process of "value generalization" (Joas 2013). On the contrary, without religions, HR accentuate their Western character (Witte and Green 2011) because they lose those potential interlocutors capable of making a bridge between HR and cultural and religious traditions on the basis of the universalizable principles contained in both. As Hans Joas (2013, p. 7) points out, "Such religious or cultural traditions may therefore discover new areas of common ground without abandoning their unique perspectives. This is the idea behind the concept of value generalization to be discussed in this chapter, which comes from the sociological theory of social change but is applied here to a more philosophical subject matter".

Indeed, this theoretical contribution of religions to the culture of peace and HR has been increasingly valued and taken up by various institutions through documents such as the UNESCO Universal Declaration on Cultural Diversity of 2 November 2001[4], the Volga Forum Declaration (2006), the San Marino Declaration (2007), point 3.5 of the document "Living Together As Equals in Dignity" (White Paper on Intercultural Dialogue, Council of Europe)[5], or the gestation of the Alliance among Civilisations by the United Nations in 2007.

However, this dialogical potential between HR and religions cannot imply either an absolute identification between both or the reduction of one to the other. Religious values cannot become a substitute for HR, but rather a complement and contribution. Nor can religions be exclusively valued as an instrument in the service of modernity, potentially leading to what Žižek has called the eventual "vampirization" of religions by post-secularization. HR, in this sense, is configured as the prerequisite for the construction of an "ethics of minimums" (Cortina Orts 2007; Sachedina 2009), which advocates respect for human dignity on the basis of justice and provides a common ground for dialogue between different understandings of the world (Ignatieff 2018, p. 26). Undoubtedly, religions form an important part of these worldviews (Laborde 2017) that can dialogue on the basis of HR. But they also have their own postulates inspired by their religious premises, which in different sociological and historical contexts and depending on the hermeneutics applied, can potentially include positions as far-ranging as the opposition to some of these rights or the struggle for a just social order that goes beyond and complements the framework of HR through demands for justice and efforts to extend them to the many people who are not yet real subjects of HR (Santos 2014). It is ultimately in this possibility of convergence without full identification between HR and religions that the complexities and ambiguities of the polygenesis and polyvalence of HR lie, but also from which the challenges of the present and the potential contributions of secular and religious actors emerge.

## 4. Conclusions: Religions and Human Rights—From the Debate on the Past to the Shared Challenges of the Present

In this article, I have explored the dialogue between two of the major languages of global humanity: the secular language of Human Rights and the language of religions (Ignatieff 2018, p. 26). In doing so, I have drawn on the current academic literature on the subject, adopting a post-secular approach to the question posed. Advances in the debate and the post-secular perspective have provided us with two keys for exploration: the continuing dialogue between the polygenesis and polyvalence of Human Rights—with important implications for framing the interplay between religions and secularity in the roots and applications of HR—and the emphasis that religions can make important contributions to the achievement of a democratic culture and HR at the global level.

From these keys, in this article, I have traced the potential shared substratum of HR and religions. Despite the complexities and difficulties in approaching this dialogue between religions and HR, as we pointed out with the debate on the Axial Age, this theoretical exercise can play an important role in rethinking our understanding of secularization, in grounding dialogue between religions, and in generating a framework of cooperation with religions for democratic culture and HR. In a certain sense, and without denying the ambiguities in the search for the possible religious roots of HR, the simple fact of theoretically positing the polygenesis of fundamental rights has important consequences

for promoting their polyvalence at the global level. In this way, the different reflections collected throughout this text have been illuminating the path of possibilities and challenges that are opening for further academic enquiry and social dialogue on the controversial relationship between Human Rights and religions.

Firstly, beyond the difficulty of clearly establishing a religious genealogy of HR and modernity, the efforts made in recent decades to explore this potential interaction on a theoretical level have favored the search for spaces for dialogue and debate, marking an alternative route to an academic reflection that has sometimes privileged the simultaneous and confrontational rather than intertwined and complex histories between secularity and religions. The quest for a common symbolic substratum (Ferry and Gauchet 2007, p. X) enhances the possibility of exploring and constructing languages, both academic and social, that serve as a meeting point between religions and HR on the shared basis of civic (An-Na'im 2011, p. 68), public and symbolic (Fernández Vallina 2011) reason, thus following in the wake already heralded by the Frankfurt School. These explorations simultaneously open the way for new research on post-secular dialogue spaces between the secular language of modernity, HR, and religions.

Secondly, the analysis of the recent literature also shows us a conviction usually forgotten in this debate: the importance of the everyday level for analyzing the realities between religions and HR. This has led to the consequent need to generate further research that, together with the development of theoretical exploration in parallel, pays more attention to the everyday level, to the realm that in religious studies has been called "lived religion" (Ammerman 2013). This can provide a space for further exploration of the possibilities and limitations of religions' actual contributions to HR.

Finally, while it is clear that the attempt to study the polygenesis of HR is subject to contradictions and ambiguities, its possibilities at the social and normative level (both for its interpretative polyvalence and for the effective religious contributions in the present day) exceed the limits of academic analysis, as we have emphasized throughout the article when discussing the possibilities that the discussion of universalizable principles and the common substrate between religions and HR offer for dialogue between religions and secular actors.

It is certainly accurate to conclude that from a socio-historical perspective, as Joas (2013) points out, it is difficult to make a precise judgement on the role of religion in HR. However, the simple recognition that the genealogy of fundamental rights may have religious roots, and that religious traditions may contribute in the present to the global and postcolonial extension of HR on the basis of universalizable principles common to both, can and has encouraged the emergence of new interpretations of the legacies and semantic potentials of religions to embrace and empower HR. In the words of Karen Armstrong (2020, p. 495): "In the past, sacred scripture did not slavishly return *ad fontes*, but always moved creatively forward to meet new challenges. Unless our traditions can address this urgent need, they will not overcome the problems of our time". In short, in this relationship between HR and religions, the analysis of their genealogy and the contribution of religions to HR in the present can go hand in hand, because it is precisely in this recognition of a past that, without denying the contradictions, is known to be shared and that the common and present challenge of defending and promoting HR can be built on the basis of cooperation between religious and secular actors.

**Funding:** This research received no external funding.

**Data Availability Statement:** Publicly available datasets were analyzed in this study. This data can be found in the sources listed below:

- https://www.fbbva.es/wp-content/uploads/2019/10/Presentacion_Estudio_Valores_Esfera_Privada_2019.pdf (accessed on 23 October 2023).
- https://www.pewforum.org/2018/05/29/being-christian-in-western-europe/ (accessed on 23 October 2023).

- https://www.pewresearch.org/global/2020/07/20/the-global-god-divide/ (accessed on 23 October 2023).

**Conflicts of Interest:** The author declares no conflict of interest.

## Notes

1    It is worth noting, in this respect, that the controversy generated at the beginning of the 21st century by the question of whether or not to incorporate Christian roots in the ultimately unsuccessful project of "Treaty establishing a Constitution for Europe". Cf. https://www.aceprensa.com/religion/la-referencia-al-cristianismo-en-la-constituci-n-e/ (accessed on 23 October 2023).

2    Cf. https://charterforcompassion.org/who-we-are/ethics-principles.html (accessed on 23 October 2023).

3    The question "Can you tell me how much you agree or disagree with the statement 'Religions today are more a source of conflict than of peace'?" was answered by the respondent on a scale of 0 ("completely disagree") to 10 ("completely agree"). The results provide a score of 6.3 for Spain, 7.3 for France, 7.1 for Germany, 7 for the United Kingdom, and 6.8 for Italy. The empirical information was obtained through a survey of a representative sample of 1,500 people, aged 18 and over, in each of the five most populated countries of the European Union.

4    Accessible online at https://en.unesco.org/about-us/legal-affairs/unesco-universal-declaration-cultural-diversity (accessed on 23 October 2023).

5    Accessible online at https://www.coe.int/t/dg4/intercultural/source/white%20paper_final_revised_en.pdf (accessed on 23 October 2023).

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
