# Peer review of "Human Rights and Religions: An Overview on a Controversial Relationship"

_religions, doi:10.3390/rel14111343_

Round 1
Reviewer 1 Report
This paper reviews mainly philosophical and sociological studies about the influence of religions on the genesis of HR, considering it as plurality of genealogies of HR and modernity (or "Western culture"), emphasised in other faiths beyond Christianity.
Recommended historical precision:
(line 165) It would be better to adapt the paragraph with a corollary explaining the following facts, and at the same time contrasting the Duranti citation, in the sense that a polyvalence is clear but a polygenesis not: only theoretically, other religions admit the equal dignity of every human creature, whereas Christianity defends it for everyone, baptized or not. It is difficult to concliliate the universality and effective recognition of rights when the main regard to the respective worshipers as persons is described, respectively, as chosen people (Hebrews) and submission (=Islam). Here follows a guideline for the suggested idea:
Human inalienable dignity at the core: why do we have it? Where it comes from? Everyone deserves the same due respect because all are children of God, created as “imago Dei” and therefore brothers and sisters, whereas secular conception does not provide with any foundation but only the premise “etsi Deus no daretur”. So, both are Western products, but one comes first in History and the other has got the contemporary hegemony. However, only within an environment that already recognizes certain rights and freedoms it is possible to develop the idea of such rights and freedoms detached from their foundations. And only thus is possible to admit a religious conversion as a free and individual decision and not as a crime that deprives the individual of the category of an equal.
Proposal for an overall concretion that would provide the text with precision and innovative perspective:
Real polygenesis should be stressed: not different religions historically combined, but simply the dialogue between them and the secularity as a genesis of the Universal Declaration of Human Rights. So, every mention to HR polygenesis or genealogy should be concreted as UDHR genealogy, or modern HR genealogy, or institutionalized HR genealogy. HR (if universal, inalienable and indivisible) exist before any express recognition has been adopted, as a part or manifestation of natural law that becomes positive law (written and published). They are recognized and not simply given as prerogatives by the established power.
Mistakes by line: 157 John XXXIII; 455 religionscannot
As for the references, they could be considered enough in quantity and quality for this kind of research, where an important amount of secondary sources should be reviewed following a clear methodology (it is explained in the introduction, based on some keywords an areas of knowledge). More than ten of the works have been published in the last five years, within a total of sixty seven.
Author Response
Dear reviewer,
Many thanks for your comments, which have been of great relevance and useful in revising and making major changes to the structure and content of my text.
You will see how I have responded to the issues you raised in the many changes introduced in the new version of the text, which I have incorporated for your convenience through the " track changes ".
Throughout the new version, you will find how I have given greater centrality to the differences between polygenesis and polyvalence, particularly in the introduction and conclusions. This has also provided the text with greater internal coherence and originality, as well as allowing us to give more relevance to the religious contributions to human rights.
I have also incorporated the specific clarification you made about the particularity of Christianity in this history of human rights, specifically in lines 68-70, 177-184 and 260-270, as well as the emphasis on the temporal primacy of Christianity in Western history, as a basis for the later culture of human rights.
The two errata noted (line 157 and 455) have been corrected.
Thank you very much again for your comments and review, and I hope that the new version will be able to satisfy all the issues you have raised.
Reviewer 2 Report
Human Rights and religions: an overview on a controversial relationship
This article aims to examine the ‘controversial’ relationship between human rights and religions – or rather, as the article evolves, between religious values, norms, or ‘languages’ and human rights. The argument is framed around two major points:
First, that the modern human rights framework is secular.
Secondly, that ‘religions’ can foster peace, democracy, harmony between ethnic groups through a human rights approach.
The article is framed at a high level of generality which for this reader poses some problems:
First, the supposed contradiction between human rights and religions is based on an assumption that there needs to be a ‘religious geneaology’ of human rights. The author refers to the ‘ambiguity of the presence of religions in the construction of the Universal Declaration of Human Rights’ (UDHR) and to the lack of reference to God in the opening document; it suggests that other ‘charters’ reference a deity.
This is incorrect in a number of ways. First it was a Declaration not a Charter; secondly the 1945 UN Charter sets a pattern for promoting human rights ‘to reaffirm faith in fundamental human rights, in the dignity and worth of the human person’ …. . After a half century of conflict, including wars based on religious discriminations, the UDHR was concerned with getting right the balance between the individual and the state. The essential conflict in the debates on the UDHR were about that relationship; according to Mary Ann Glendon, A World Made New: Eleanor Roosevelt and the UDHR (2001) the human rights project was ‘peripheral’. The main debates were about political philosophy, but religious and philosophical views were not ignored. UNESCO established a Committee of Experts to survey the philosophical foundations of human rights. The participants included those of different cultures and faiths as did the participants in the debates on the UDHR.
Moreover, references to a religion or deity are found in some national constitutions but not in international law instruments.
To look for the ‘religious geneaology’ of human rights the author needs to look back to the natural law debate. It is not clear why there needs to be a ‘religious geneaology’ of human rights
The argument about religions on the other hand is based on the academic literature and the search for common values in various religions. The argument on this point makes more sense but I think the author misses the point that the contradictions lie within the ‘religions’ themselves. It is a fact that ‘faith-based’ institutions are the strongest supporters of social justice issues in many communities in many countries, but at the same time ‘religions’ are a source of human rights abuses; for example discrimination on religious grounds within a country (such as Pakistan) or discrimination on the basis of gender or ethnicity in many theocratic states.
As I mentioned, the article is framed at a high level of generality and in my opinion this makes the analysis less coherent than it might be. For example, the use of the word ‘categories’ of ‘religions’ and human rights begs the question: ‘categories’ of what? The individual’s right to freedom of religion is recognised in the UDHR as an individual human right which stems from the right to freedom of thought, and as a collective right. The individual’s right to freedom of religion also includes the right to practice, to manifest one’s beliefs. It can be an individual or a collective right. See UDHR Art 18, ICCPR Arts 18, 19.
However, the author is often referring to ‘religions’ as ‘collectives’ of practices and sometimes as an individual right. These ‘rights’ have different meanings.
As the article proceeds, it is clear that the focus is on the ‘terms’ or ‘languages’ and ‘dialogue’ or debate around religious norms. The reference to Habermas shows the genesis of the argument in the article. I think this should be made much clearer at the beginning of the article. I did not find it easy to follow the argument.
I do not think the overall argument is deficient (at least in relation to ‘religions’), it is rather that way it proceeds and is structured. I do however think that the author expects too much of the UDHR and oversimplifies the argument about the ‘religious genealogy’ of human rights.
Author Response
Dear reviewer,
Many thanks for your comments, which have been of great relevance and useful in revising and making major changes to the structure and content of my text.
You will see how I have responded to the issues you raised in the many changes introduced in the new version of the text, which I have incorporated for your convenience through the " track changes " tool.
Firstly, the need for a religious genealogy of human rights has been corrected and throughout the text I have opted to talk about the polygenesis of human rights.
Secondly, the information on the Universal Declaration of Human Rights has been amended, following your suggestions and including the reference to Mary Ann Glendon's work (lines 162-200).
Thirdly, the presence of the debate on natural law has been reinforced all along the text, particularly through the reference to Maritain in line 182.
Fourth, the ambiguity of the relationship between human rights and religions has been highlighted more clearly throughout the text, particularly in section 2 and in the conclusions. While the previous text already echoed this issue, following your suggestions, I have reinforced this contradictory character.
Fifth, I have specified and particularised my understanding of religions, which in this case focuses first and foremost on "religious tradition", which I define in line 55. Similarly, excessively generic concepts such as "categories" or "religions" have been eliminated and more precise alternatives have been sought throughout the text.
The introduction and conclusions have been revised to clarify my approach to the debate and, above all, to place greater emphasis on three concepts: polygenesis, polyvalence and effective contributions, which I think serve both to make the text more coherent and to give it a greater liveliness that can capture the contradictions between religions and human rights to which you pointed.
Thank you very much again for your comments and review, and I hope that the new version will be able to satisfy all the issues you have raised.
Reviewer 3 Report
see file, attached, for extensive comments.

Author Response
Dear reviewer,
Many thanks for your comments, which have been of great relevance and useful in revising and making major changes to the structure and content of my text.
You will see how I have responded to the issues you raised in the many changes introduced in the new version of the text, which I have incorporated for your convenience through the " track changes " tool.
Indeed, the article is in the nature of a literature review and is therefore intended as an introduction to the debates on the subject that have taken place in recent years, with particular attention to the works of Joas and Witte and Green, which are frequently cited in other reflections of a similar kind. However, in order to further increase the number of sources in the article, three additional references have been added, as well as reinforcing the arguments derived from the logic of the text and giving it a more independent character, as you will see in the added reflections on polygenesis - polyvalence and effective contribution.
Missing references in the final bibliography have been incorporated.
It has been more explicitly clarified that my approach to the debate is post-secular, although throughout the article the weight between religions and secularity has been balanced in order to show a more complex and not so unidirectional approach. In this respect, the main ideas about the Axial Age have been clarified, specifically what you pointed out to me about the inclusion of Greek authors and Islam within this category (lines 243-254) , and the historical part has been approached with a greater emphasis on the contradiction and POTENTIAL polygenesis of human rights that can be observed in this principle of generalisation of values, emphasising that it is a contradictory rather than an undisputed issue (lines 309-322).
This is one of the main changes made throughout the article, which, without renouncing the hypothesis of the polygenesis of human rights (not so much of the declaration, but rather of its conceptualisation), the current version shows more contradiction and ambiguity in order to reflect an equally complex and controversial history in this regard. In this vein, in-depth corrections have been made in section 2 of the article, as well as in the last part of the introduction and in the conclusions.
Likewise, the idea of a common ethic has been corrected, and instead in the current versión I speak in of a hermeneutic that is capable of constructing a dialogical space based on the different religious traditions (line 330)., 360-368).
Thank you very much again for your comments and review, and I hope that the new version will be able to satisfy all the issues you have raised.
Round 2
Reviewer 2 Report
This article is much improved after taking into account my previous comments.
I have just a couple of quibbles:
First, after referring to human rights and religions as ‘concepts’ \ ‘categories’ in the abstract, the author uses the term ‘words’ in the first sentence of the article. This is dissonant – I suggest to use ‘concepts’ or ‘categories of organisation of knowledge’ or similar. Further I think this first sentence is awkwardly expressed. I would suggest a rewording along the lines of:
‘HR and religions are two categories that are commonly associated and intertwined … ‘
For similar reasons, on p2, line 44 ‘words’ can be deleted without changing the sense.
Congratulations - it's a great piece.
N\A
Author Response
Thank you very much for your evaluation of the text and for your helpful comments on its content.
Following your recommendations, I have changed "words" in the first sentence to "categories of organisation of knowledge", and I have rephrased the beginning of the article, in line with your advice.
I have also deleted "words" in line 44.
Once again, thank you very much for all your suggestions. Thanks to your criticisms and comments, I do believe that the text has indeed been improved.